# Anti-Platelet Aggregation and Anti-Cyclooxygenase Activities for a Range of Coffee Extracts (*Coffea arabica*)

**DOI:** 10.3390/molecules26010010

**Published:** 2020-12-22

**Authors:** Nuntouchaporn Hutachok, Pongsak Angkasith, Chaiwat Chumpun, Suthat Fucharoen, Ian J. Mackie, John B. Porter, Somdet Srichairatanakool

**Affiliations:** 1Department of Biochemistry, Faculty of Medicine, Chiang Mai University, Chiang Mai 50200, Thailand; n.hutachok@hotmail.com; 2Royal Project Foundation, Chiang Mai 50200, Thailand; pongsak.a@cmu.ac.th (P.A.); ssomdet@hotmail.com (C.C.); 3Thalassemia Research Center, Institute of Molecular Biosciences, Mahidol University Salaya Campus, Nakornpathom 71300, Thailand; suthat.fuc@mahidol.ac.th; 4Haemostasis Research Unit, Department of Haematology, University College London Medical School, London WC1E 6BT, UK; i.mackie@ucl.ac.uk; 5Red Cell Disorder Unit, Department of Haematology, University College London Medical School, London WC1E 6BT, UK; j.porter@ucl.ac.uk

**Keywords:** chlorogenic acid, coffee, cyclooxygenase, espresso, instant coffee, platelet aggregation

## Abstract

Coffee is rich in caffeine (CF), chlorogenic acid (CGA) and phenolics. Differing types of coffee beverages and brewing procedures may result in differences in total phenolic contents (TPC) and biological activities. Inflammation and increases of platelet activation and aggregation can lead to thrombosis. We focused on determining the chemical composition, antioxidant activity and inhibitory effects on agonist-induced platelet aggregation and cyclooxygenase (COX) of coffee beverages in relation to their preparation method. We prepared instant coffee and brewed coffee beverages using drip, espresso, and boiling techniques. Coffee extracts were assayed for their CF and CGA contents using HPLC, TPC using colorimetry, platelet aggregation with an aggregometer, and COX activity using ELISA. The findings have shown all coffee extracts, except the decaffeinated types, contained nearly equal amounts of CF, CGA, and TPC. Inhibitory effects of coffee extracts on platelet aggregation differed depending on the activation pathways induced by different agonists. All espresso, drip and boiled coffee extracts caused dose dependent inhibition of platelet aggregation induced by ADP, collagen, epinephrine, and arachidonic acid (ARA). The most marked inhibition was seen at low doses of collagen or ARA. Espresso and drip extracts inhibited collagen-induced platelet aggregation more than purified caffeine or CGA. Espresso, boiled and drip coffee extracts were also a more potent inhibitors of COX-1 and COX-2 than purified caffeine or CGA. We conclude that inhibition of platelet aggregation and COX-1 and COX-2 may contribute to anti-platelet and anti-inflammatory effects of espresso and drip coffee extracts.

## 1. Introduction

Arabica coffee (*Coffea arabica*) and Robusta coffee (*Coffea canephora*) are the most widely-consumed coffee beverages in the world. These coffee varieties differ in terms of cultivation area, their physical and chemical characteristics and the taste of the resulting coffee beverage [1]. Coffee beans contain lipids, carbohydrates, minerals, caffeine (CF) and phenolic acids such as chlorogenic acid (CGA) or 5-*O*-caffeoylquinic acid (CQA), caffeic acid (CA), ferulic acid (FA), quinic acid (QA), and feruloylquinic acid (FQA) (Figure 1), all of which contribute to their biological and pharmacological effects [2].

Boiled coffee is the earliest and most commonly prepared form of coffee brew and is commonly referred to as Turkish coffee. Espresso coffee is a complex and much appreciated beverage among coffee consumers in Southern Europe, Central America and other areas. Indeed, differing varieties of coffee drink preparations will result in differences in aromatic compositions, bioactive compounds such as CF, CGA, CA, and other phenolic compounds, as well as other resulting pharmacological properties.

Importantly, coffee consumption provides hepato- and cardio-protective effects, displays antimicrobial activity and lowers the risk of type 2 diabetes mellitus [2]. In healthy people, moderate caffeine consumption (400 mg/day) neither induces cardiac arrhythmia nor increases serum cholesterol levels [3]. However, excessive coffee consumption is known to result in mineral deficiency, hypertension, hypercholesterolemia, insomnia, tremors, nausea, polyuria, diarrhea, and polyphagia [2,3,4,5,6]. Possibly, the positive effects of coffee are caused by certain compounds such as CF and CGA [3,7]. CGA (or caffeoylquinic acid, CQA) together with feruloylquinic (FQA) and dicaffeoylquinic (diCQA) acids are phenolic acids [7,8,9], and CF is recognized as a remarkable alkaloid [2,3]. These compounds are notably predominant in green coffee beans. Interestingly, the contents of these compounds in coffee are varied among tree species and can vary according to processing techniques [2,10,11,12,13], both of which can lead to differing degrees of antioxidant and radical scavenging activities. Nevertheless, degradation of CGA during the roasting process did not ultimately affect the antioxidant activity of the coffee beverage [14].

The hypercoagulable state is one of the most common hematological complications that can be caused by platelet activation, red cell disorders, the presence of reactive oxygen species (ROS) and/or decreased natural anticoagulants [15,16,17]. Regular blood transfusions and iron chelation therapy have become the main prophylactic methods employed in haemoglobinopathies (commonly in β-thalassaemia intermedia patients) to eliminate these multi-factors and prevent secondary complications [18,19]. In addition, platelet antagonists such as eptifibatide and prasugrel were found to reduce platelet activation and aggregation in patients with sickle cell diseases [20,21]. Notably, coffee is known to exhibit anti-thrombotic, nitric oxide (NO^●^) modulating and anti-platelet effects [22]. Accordingly, high and chronic consumption habits of coffee could alter the response of platelets to adenosine actions, decrease platelet aggregation and lower the risk of cardiovascular disease [23]. Moreover, cyclooxygenases (COXs), which are identified as constitutive COX-1 and inducible COX-2, can catalyze the synthesis of prostaglandins (PGs) and thromboanes (TXs) and are inhibited by acetylsalicylic acid (ASA). Specifically, COX-2 inhibitor drugs inhibit the synthesis of platelet-inhibitory prostacyclin (PGI_2_); nonetheless, NS-398 was found to induce certain side effects such as increased vascular platelet adhesion, leading to thrombosis and hypercoagulation in patients with cardiovascular diseases [24]. Notably, they have been reported as being able to promote atherothrombosis by inhibiting the vascular formation of PGI_2_ and increasing thrombotic-associated risks [25]. We have hypothesized that coffee preparations with different amounts of CGA and CF could help to prevent a hypercoagulable state, by inhibiting platelet aggregation and COXs and thereby, assist in the inhibition of platelet aggregation and COX activities to differing degrees. The aims of this study were to prepare different roasted Arabica coffee drinks using boiled (brewed coffee) and pressurized (espresso beverage) extraction methods and then evaluate their inhibitory effects on platelet aggregation and COXs activity.

## 2. Results

### 2.1. Chemical Compositions and Antioxidant Activity in Coffee Preparations

HPLC/DAD analysis revealed that authentic 125 μg/mL caffeine (CF), 125 μg/mL chlorogenic acid (CGA) and 1 mg/mL caffeic acid (CA) were eluted at the specific retention time (T_R_) values of 11.72 min, 12.50 and 17.18 min, respectively (Figure 2). Consistently, free forms of CF and CGA at the corresponding time points were almost exclusively detected in extracts of instant coffee (both regular and decaffeinated formulas), as well as in extracts of espresso, drip and boiled coffees. However, CA was notably not detectable. In addition, the chromatographic profiles of CF and CGA were very similar among these coffee extracts with the exception of the decaffeinated coffee. Not surprisingly, nearly all of the CF was eliminated from the decaffeinated instant coffee. With regard to espresso, drip, boiled and instant (regular) coffee extracts, the peaks eluted between 2 and 10 min were observed to be evenly predominant. This outcome was possibly attributed to the presence of other phenolic/flavonoid compounds.

Results presented in Table 1 reveal differences of CGA, CF, TPC, and AA contents among five coffee extract samples, among which the CF and CGA contents are notable. Decaffeinated coffee is usually prepared by employing a treatment using an organic solvent or carbon dioxide to remove intact caffeine from regular coffee, reaching 1–2% of the original content of the coffee. Herein, instant decaffeinated coffee was found to contain higher CGA content but less CF content than instant regular coffee. Additionally, it displayed higher CGA, TPC, and antioxidant activity than other coffee extracts.

### 2.2. Inhibitory Effect of Coffee Extracts on Platelet Aggregation

In this study, the platelets present in platelet rich plasma (PRP), were treated with different coffee extract preparations before adding different platelet agonists in vitro and monitoring platelet aggregation using an aggregometer instrument. All espresso, drip and boiled coffee extracts tended to cause dose dependent inhibition of platelet aggregation induced by ADP, collagen, epinephrine, and arachidonic acid (ARA), in which the espresso and drip coffee extract at the dose of 1 mg/mL showed significant inhibition of platelet aggregation induced by ADP (5 μM) and collagen (1 μg/mL). However, the inhibitory effects became weaker when the doses of the agonists were increased. The most marked inhibition was seen with low doses of collagen and with ARA. In addition, the inhibitory effects of the coffee extracts on the platelet aggregation to U46619 were weak (only occurring in 1/5 donors) and there was no inhibition of aggregation to TRAP (Figure 3). These results indicate that the inhibitory effects of coffee extracts on platelet aggregation differ depending on the particular pathways and mechanisms leading to platelet aggregation that are activated by different agonists.

As shown in Figure 4, CF (54.17 mg/mL) was found to be more effective than CGA (18.23 mg/mL) in inhibiting the aggregation of platelets induced by collagen. However, they showed less inhibitory effects on platelet aggregation than the coffee extracts, even though their concentrations were slightly higher than those of espresso (48.83 μg CF and 8.60 μg CGA equivalent/mL), boiled coffee (49.58 μg CF and 9.45 μg CGA equivalent/mL), drip coffee (45.73 μg CF and 8.81 μg CGA equivalent/mL), instant coffee (39.81 μg CF and 5.15 μg CGA equivalent/mL), and decaffeinated instant coffee (0.53 μg CF and 10.61 μg CGA equivalent/mL). There were no clear differences in the degree of inhibition between the coffee types at equal concentrations (1 mg/mL). Similarly, CF and CGA slightly inhibited the aggregation of platelets induced by collagen and showed much lower inhibitory effects than the coffee extracts. Therefore, the inhibitory effects of Arabica coffee and instant coffee on platelet aggregation were interesting, but will require further investigation.

In time course experiments, coffee extracts (1 mg/mL) were mixed with PRP in the cuvette and incubated for indicated periods of time. Collagen and ARA agonist were then added and platelet aggregation were immediately measured using an aggregometer instrument. Espresso coffee slowly inhibited platelet aggregation while boiled and drip coffee extracts caused immediate (within 1 min) inhibition of platelet aggregation to collagen (1 µg/mL). Likewise, the degree of inhibition was more potent at higher concentrations and over longer incubation periods. Consistently, the degree of inhibition fell in the following order: Boiled coffee ~ drip coffee > espresso coffee (Figure 5A–C). In contrast, when using ARA (1 mM) as an agonist, all of the coffee extracts completely inhibited platelet aggregation within 1–2 min regardless of concentration. However, the degree of inhibition fell in the order of boiled coffee ~ drip coffee > espresso coffee (Figure 5D–F).

Moreover, the inhibitory effects on the agonists-induced platelet aggregation seemed to be dependent upon the incubation time of the coffee extracts. Consequently, the inhibition potency was found to be in the order of drip coffee > boiled coffee > espresso coffee with regard to ARA stimulation, whereas it was not observed to be different with regard to collagen stimulation (Figure 6A–B).

### 2.3. Effect of Coffee Extracts and Instant Coffee on Cyclooxygenase Activities

Treatments with espresso, boiled and drip coffee extracts (2 mg/mL equivalent to 97.7, 99.2, and 91.4 μg CF, 17.2, 18.9, and 17.6 μg CGA, respectively) and indomethacin (100 mg/mL) considerably inhibited COX-1 activity (*p* < 0.05), while CF (100 μg) and CGA (20 μg) treatments were found to be less active. Similar to COX-1 inhibition, the coffee extracts revealed significant inhibitory effects on COX-2 activity that were noticeably weaker than indomethacin (100 mg/mL) but stronger than CF and CGA (Figure 7). The inhibitory effects of the three coffee extracts compared with CF and CGA on COX-1 and COX-2 are broadly similar. The results suggest that with respect to inhibition of COX-1 and COX-2, that there are compounds present in coffee extracts, which are relatively lacking in CGA or CF. Although these differences appear marked for both COX-1 and COX-2 (Figure 7), further experiments would be required to establish statistical significance under a range of conditions.

As is shown in Figure 8, the results demonstrate that espresso, boiled and drip coffee extracts exerted almost equal degrees of inhibition of COX-1 and COX-2 activities in a concentration-dependent manner, while CF and CGA were found to have exerted lesser degrees of inhibition. In the case of a constitutive COX-1, the degree of inhibition was minimal at a dose of 0.15 mg/mL and considerable at doses of 1–2 mg/mL. In the case of an inducible COX-2, the inhibitions were increasingly obvious as the doses were increased. Even with less CF and CGA equivalent concentrations, all the coffee extracts still displayed more potent degree of inhibition of COX-1 and COX-2 activities than the authentic standards.

In this study, the coffee extracts are likely to contain other bioactive phenolic/flavonoid compounds (such as QA, FA, and FQA) besides CF and CGA that have synergistic anti-platelet aggregation and health benefits.

## 3. Discussion

Natural bioactive compounds in coffee beans help promote human health and wellness; however, the varying classes of phenolic compounds, sources of antioxidants, extraction methods, thermal processing and sample storage may influence the biological and pharmacological effects of coffee drinks. Generally, instant coffee is prepared by extracting the soluble and volatile contents of coffee beans or pulp with pressurized hot water (approximately 175 °C) and then through varying degree of concentration using either evaporation or spray drying methods of preparation. Instant coffee has been reported to possess anti-oxidative and anti-peroxidation properties; however, it may also act as a pro-oxidant at high concentrations, probably by converting an inactive ferric ion (Fe^3+^) to a redox-active ferrous ion (Fe^2+^) in a Fenton reaction [26]. Importantly, coffee contains significantly higher amounts of free phenolic compounds and displays greater antioxidant activity levels than cocoa and tea, of which their anti-oxidation yields are highly related to their TPC content; particularly with regard to their free phenolic compound content. This outcome is possibly due to the presence of the main content of CGA, particularly 5-caffeoylquinic acid (5-CQA) [27]. Natella et al. [28] reported on the presence of CGA, but not of CA, *p*-coumaric acid (CMA) and ferulic acid (FA) in American-style filtered coffee (60 g roasted coffee/L water) that had been prepared from a commercial automatic brewing machine. On the contrary, CA, CMA, and FA, but not CGA, were detectable in the brewed coffee after alkaline hydrolysis. In the present study, we found only the free form CGA, but not CA, with regard to the authentic standards in all of our coffee extracts. This outcome suggests that most or all of the CA in coffee would be esterified with QA and eventually form caffeoylquinic acid or CGA. In terms of roasting, 4-vinylcatechol of the CA moieties in green coffee usually oligomerizes to form bitter compounds [29].

The extraction phase of espresso coffee combines physical and chemical variables in a very short period of time. These variables directly affect its flavor, aroma, quality, and the resulting coffee beverage’s bioactivities [30]. When compared with green coffee beans, roasted coffee beans are known to contain considerably lower contents of phytochemicals. This was confirmed by a 68% decrease of CGA. However, in this study, higher TPC and AA contents and unchanged CF contents were recorded [31]. Additionally, secondary metabolites, particularly CGAs, that are present in green beans were found to be degraded during the roasting process into other phenolic compounds. This is ultimately responsible for the bitterness of the coffee beverage. The present findings have revealed that all the coffee extracts exhibited differences in TPC and AA as a result of the differing coffee processes. Consistently, the TPC and AA levels detected in instant decaffeinated coffee and espresso coffee beverage were directly related to the CGA content, but were inversely related to CF content. Moreover, technological factors such as those associated with the decaffeination process, the brew volume and the method of preparation test, as well as the coffee species and the degree of roast, may all affect antioxidant activity and phenolic content in coffee drinks. Aguilera and colleagues have reported that heat/acidity-assisted extraction of coffee has produced higher concentrations of CGA, vanillic acid (VA), protocatechuic acid (PCA) and CMA than conventional methods [32]. In addition, abundant contents of CF (approximately 1200 ± 57 mg/kg spent coffee residual) and CGA (approximately 1700 ± 90 mg/kg spent coffee residual), some phenolic acids (e.g., CA, FA, PCA, VA, gallic acid, syringic acid and *trans*-cinnamic acid), and some flavonoids (e.g., rutin, cyanidin 3-glucoside and quercetin) were also present in the final coffee beverage [33]. The findings of Aguilera and colleagues have supported those of our study by confirming greater TPC and AA levels in the decaffeinated coffee than in the coffee brew extractions. With regard to free radical-scavenging activity and TPC, espresso (decaffeinated) coffee (30 mL cup) displayed significantly greater effects than espresso (regular), while a long-extraction espresso coffee beverage (70 mL cup) was found to result in greater radical-scavenging activity when compared to a short-extraction espresso coffee beverage (20 mL cup). Furthermore, Robusta brewed coffee revealed significantly greater radical-scavenging activity than Arabica brewed coffee [34]. In addition, at least 23 hydroxycinnamic derivatives including CGA, lactones and cinnamoyl-amino acid conjugates were detected in the coffee brews [34].

Thermal processing (such as steaming, boiling or frying) helps release bound phenolic compounds from the structural matrices and glycosides in coffee beans, but tends to reduce antioxidant activity. However, high-pressure processing tends to better retain the antioxidant properties of coffee drinks when compared to thermal treatments [35]. CGA, CA and FA are the three main anti-oxidative phenolic compounds present in both roasted coffee grounds and spent coffee grounds. Of these, CGA is the most abundant compound [36]. In the present study, CA was not detectable in all coffee extracts, while authentic CA was detected in the chromatogram. In modifications, feruloyl esterase obtained from microbial substances may be included in the preparation of coffee to release anti-oxidative phenolic acids such as CA, PCA, and FA from the coffee pulp and beans and would then contribute in the way of health benefits [37]. Moreover, water coffee extracts can be prepared using manual methods (such as boiling) and automatic coffee makers (such as percolation, espresso, and drip) depending upon the facility used and the customers’ taste and degree of satisfaction. According to both the aroma and flavor, roasted coffee drinks are extremely popular when compared to both green coffee and instant coffee; nonetheless, roasting may influence the nutraceutical, biological, and pharmacological properties of the coffee beverage. Interestingly, medium roasted coffee displayed the highest TPC content, while light roast coffee exhibited the highest 2,2-diphenyl-1-picrylhydrazyl (DPPH) radical scavenging activity. Lastly, dark roasted coffee revealed both the lowest TPC content and the lowest degree of radical-scavenging activity [38]. We found that coffee extracts were capable of inhibiting platelet aggregation induced by ADP, epinephrine, collagen, and ARA, for which the most remarkable degree of inhibition took place with the ARA agonist, while inhibition was more marked at lower doses of collagen (which are thought to activate platelets mainly through the cyclo-oxygenase pathway). According to the results, high inter-individual variability, coupled with the fact that the sensitivity to inhibition is influenced by the strength of the aggregation response in the presence of PBS in each individual (i.e., if the response is at the lower end of normal, there is a more limited scope for inhibition than if the response was upper normal, since aggregation responses below zero cannot be recorded). Additionally, Disaggregation is thought to occur when the level of stimulus is below a certain threshold for full, irreversible aggregation, i.e., only limited amounts of TXA2 are generated and are compensated for by the inhibitory pathways within the platelet (e.g., cAMP generation). The concentration of TXA2 is also likely to be insufficient to provoke ADP release from the dense granules. Surprisingly, we found that inhibition was not influenced by CGA and CF when compared to the standard. Importantly, CF augmented the activity of NSAIDs such as aspirin, indomethacin and ketoprofen; nevertheless, it did not influence NSAIDs-induced inhibition of platelet thromboxanes (TXs) [39]. Likewise, there is increasing evidence that the consumption of phenolic-rich beverages has the potential to lower and prevent the risk of developing thrombosis. For example, catechol, hydroquinone and their derivatives, which are the antioxidants present in coffee and other plant products, were found to inhibit the production of prostaglandin E_3_ (PGE_3_) and TXB_2_, as well as the aggregation of platelets induced by ARA but not by U46619 [40,41]. In addition, CGA has displayed inhibitory effects on collagen-induced platelet aggregation and TXA_2_ production in a concentration dependent manner, along with increases in microsomal platelet cyclic adenosine-5′-monophosphate (cAMP) and cyclic guanosine-5′-monophosphate (cGMP) levels [42]. In contrast, CA was not detectable in all brewed coffee and instant coffee samples in this study, even though it had been previously identified as an intense bitter compound in strongly roasted espresso coffee [29]. With regard to CA composition, the brewed and instant coffee beverages displayed proper inhibitory effects on platelet aggregation and COXs activities. This was possibly due to the actions of other bioactive compounds. Interestingly, espresso coffee contains the highest amount of CGAs, mainly 5-CQA, (approximately 5 mM) and much lower amount of CA among all commonly consumed beverages [43]. With regard to its bioavailability, CGAs per se may directly interact with platelets in blood circulation or can be metabolized extensively by colonic microbiota into degradation products including CA, FA, hydrocaffeic acid, dihydroferulic acid and 3-(3′-hydroxyphenyl)propionic acid before being absorbed from the gut lumen [44]. Nevertheless, consumption of instant coffee (16 g/d, equivalent to 520 mg CF) for a 3-wk period did not change the plasma and urinary levels of TXB_2_ and PGE_2_ [45]. Furthermore, antioxidant activity of coffee products has been reported in the order of Robusta coffee, coffee cherry, Indian green coffee (72%) > CA (71%) > CGAs (70%) > dicaffeoylquinic acids (69%) > Nescafe Espresso (49%) [46].

Lastly, we have identified the mechanism of the coffee extracts on ARA-induced platelet aggregation by estimating the degree of COX activity through quantitation of prostaglandin. We found that the extracts were more potent than CGA or CF alone in inhibiting the aggregation by suppressing both COX-1 and COX-2 in the PGE_2_ pathway; which the inhibition would possibly be interactions of CGA with CF. The two compounds with other phenolic compounds (such as quercetin), demonstrating the synergistic effects on inhibiting pro-inflammatory responses. Similarly, combination of quercetin and tea saponin extract exerted synergistic effect on inhibiting PGE_2_ production through the COXs/PGE_2_ pathway [47]. Naito and colleagues have previously demonstrated that hot water extracts of Blue Mountain, Yunnan and Kilimanjaro coffee beans containing CGA, CF, QA, and CA exhibited weak anti-platelet aggregation activity [48]. Moreover, CGA (5 mg/kg) and CF (0.5 g/kg) treatment nearly 60% decreased platelet aggregation in streptozotocin-induced diabetic rats [49,50]. Nonetheless, decaffeinated lyophilized coffee which contained higher phenolic compounds and CGA and integral coffee beverage did not change the hemostatic parameters in normal and high fat-fed rats [51]. Interestingly, an anti-oxidative diterpene “kahweol” found in unfiltered coffee has been shown to have a significant anti-inflammatory effect on the inhibition of inducible COX-2 and nitric oxide synthase (iNOS) activities and monocyte chemoattractant protein-1secretion [52,53,54]. Surprisingly, CGA has been reported to bind to the active site of the adenosine A_2A_ receptor and consequently attenuates the anti-platelet effect. Moreover, coffee extracts and authentic CGA, GA, and kaempferol consistently suppressed the expression of the COX-2 gene in macrophage (RAW264.7-Mφ) cells [55] and in regular coffee beverages, while CF inhibited COX-2 gene expression in rats [56]. Furthermore, a recent study has reported that the anti-inflammatory effect on COX-2 gene suppression could be attributed to the extracts of the coffee leaves, which contain CGA, CF, mangiferin, rutin, and other bioactive molecules [57]. Synergistically inhibitory effect of agonists-induced platelet aggregation, which is related to antioxidant activity and reported in extracts of plants (such as *Salvia* species), is possibly caused by the interaction of persisting phenolic compounds [58,59,60]. Taken together, our findings suggest the potential inhibition of platelet aggregation and COXs activities in the following order: Drip coffee > espresso coffee > boiled coffee > instant coffee > CF, CGA, suggesting that the degree of inhibition may not be influenced by the anti-oxidative compounds and caffeine present in the coffee extracts.

## 4. Materials and Methods

### 4.1. Chemicals and Reagents

Accordingly, 2,2′-Azino-bis(3-ethylbenzothiazoline-6-sulfonic acid) diammonium salt (ABTS), Dulbecco’s minimal essential medium (DMEM), RPMI1640 medium, phosphate buffered saline pH 7.0 (PBS), fetal bovine serum (FBS) and other materials for cell culturing were purchased from Gibco, Life Technologies (Eugene, OR. USA). Histopaque^®^-1077, dimethyl sulfoxide solution (DMSO), adenosine 5′-diphosphate (ADP) sodium salt, epinephrine, sodium arachidonate (AA), Thrombin receptor agonist peptide (TRAP), 9,11-dideoxy-11α,9α-epoxymethanoprostaglandin F_2α_ (U46619), gallic acid (GA), penicillin-streptomycin, 6-hydroxy-2,5,7,8-tetramethylchroman-2-carboxylic acid (Trolox), chlorogenic acid, and caffeine were purchased from Sigma-Aldrich Chemicals Company (Gillingham, UK). Acetonitrile, phosphoric acid, Folin-Ciocalteau reagent, potassium thiosulfate (K_2_S_s_O_8_) and sodium carbonate (Na_2_CO_3_) were purchased from Merck KGaA (Darmstadt, Germany). Collagen suspension was obtained from Helena Biosciences (Gateshead, UK). Competitive enzyme linked immunosorbent assay (cELISA) kit for ovine/human cyclooxygenase (COX) (No.560131) was purchased from Cayman Chemicals Company (Ann Arbor, MI, USA). UltraPure water and other chemicals and solvents that were used in this study were of the highest analytical grade.

### 4.2. Coffee Extract Preparation

Green coffee beans (Coffea arabica) were kindly provided by the Royal Project Foundation, Chiang Mai, Thailand. Coffee beans were peeled, dried at room temperature until humidity reached <13% of their original weight, roasted in a coffee roaster at 200–250 °C for 15–20 min, ground using a milling machine, and kept in aluminum-foil bags at room temperature. Coffee extracts were prepared in coffee brewing devices using the boiled/brewed process and the pressurized espresso and drip methods according to the manufacturer’s instructions and the description established by Caprioli et al. [61]. For the boiled and brewed method, ground roasted coffee (10 g) and boiled water (100 mL, 90 °C) were combined in a cylindrical vessel. The coffee was then left to brew for a few min. A circular filter was then tightly fitted to the cylinder and a fixed plunger was pushed down from the top through the liquid to force the grounds to the extract bottom containing most of the active substances. This process makes the coffee beverage stronger and leaves the spent coffee grounds in the vessel. Finally, the boiled coffee brew was then poured from the container, and this was what was used in our experiments. By using manual drip coffee equipment (Brand Delonghi, Model BCO421.S, Treviso, Italy), boiled water (100 mL, 90 °C) was poured into a container filled with ground coffee (10 g) with a perforated filter base. The hot water was then allowed to seep for approximately 5 min (15 bar pressure) through the filter. The resulting coffee then dripped into the container below. For the espresso method, a limited amount of hot water (90 °C) was pressurized at 12 bars using a portable espresso coffee machine (Brand MINIMEX, Model Miniespresso, Bangkok, Thailand). The hot water would then percolate through the ground roasted coffee beans (10 g) in a relatively short period of time to yield a small cup of concentrated foamy coffee extract. Coffee brew extracts were filtrated through filter paper (cellulose type, Whatman’s No. 1, Merck KGaA, Darmstadt, Germany), centrifuged at 3000 rpm for 15 min, freeze-dried using a lyophilisation machine and kept at −20 °C until being analyzed and studied. In this experiment, instant (regular and decaffeinated) coffee (Tesco Gold, Tesco Stores Limited, United Kingdom) (10 g) was freshly prepared by reconstituting the coffee product with boiled water (100 mL, 90 °C) and then filtering the resulting beverage through Whatman’s No. 1 filter paper.

### 4.3. Chemical Analysis of Coffee Preparations

CGA and CF were quantified using high-performance liquid chromatography/diode array detection (HPLC/DAD) [62]. The conditions included a column (C18-type, 4.6 mm × 250 mm, 5 µm particle size, Agilent Technologies, Santa Clara, California, United States), isocratic mobile-phase solvent containing 0.2% phosphoric acid and acetonitrile (90:10, *v*/*v*), a flow rate of 1.0 mL/min and a wavelength detection of 275 nm for CF and 330 nm for CGA. Data were recorded and integrated using Millennium 32 HPLC Software. CGA and CF were identified by comparison with the specific TR of the authentic standards and determined concentrations were established from the standard curves constructed from different concentrations.

Total phenolic compounds (TPC) of the coffee extracts were determined using the Folin–Ciocalteu method [63]. Briefly, the coffee extract (100 µL) was incubated with 10% Folin–Ciocalteu reagent (200 µL) at room temperature for 4 min and then incubated with 700 mM Na2CO3 (800 µL) for 30 min. The optical density (OD) was then measured at 765 nm against the reagent blank. GA (6.25–200 µg/mL) was used to generate a calibration curve that was used to calculate TPC and identify the gallic acid equivalent (GAE).

Antioxidant activity was determined using the ABTS radical cation (ABTS^●^+) decolorization method [64]. In the assay, ten microliters of absolute ethanol (blank) and coffee extracts (62.5–4000 μg/mL) or Trolox (6.25–800 µg/mL) were incubated with a freshly prepared solution consisting of 3.5 mM ABTS^●^+ and 1.22 mM K2SsO8 in the dark at room temperature for exactly 6 min. The results were photometrically measured at 764 nm. Results were expressed as the percentage of inhibition of ABTS^●^+ production and were reported in terms of mg trolox equivalent antioxidant capacity (TEAC)/g extracts.

### 4.4. Determination of Anti-Platelet Aggregation Activity

#### 4.4.1. Isolation of Platelet-Rich Plasma

Blood collection was permitted by the director of Maharaj Nakorn Chiang Mai, Faculty of Medicine Chiang Mai University, Chiang Mai, Thailand (Reference Number 8393(8).9/436) and approved by the Research Ethics Committee for Human Study, Faculty of Medicine, Chiang Mai University, Chiang Mai, Thailand (Research ID: 7575/Study Code: BIO-2563-07575/Date of Approval: 28th September 2020). Informed consent was provided by healthy blood volunteers. Subjects were asked to avoid drinking caffeine-containing beverages for 24 h prior to blood collection. At 9.00 am, blood samples of non-fasting volunteers were collected directly from veins and deposited into tubes containing 0.106 M tri-sodium citrate. The tubes were mixed gently and centrifuged at 170 g at an ambient temperature (25 °C) for 10 min. Platelet-rich plasma (PRP) supernatant was then transferred to capped polypropylene tubes. Residual blood was centrifuged at 2000 g for 15 min and platelet-poor plasma (PPP) supernatant was separated using a plastic pipette then stored in a capped polypropylene tube for use as a PRP diluent and for setting the transmission blank on the aggregometer. PRP platelet counts were obtained using a KX-21 cell counter (Sysmex Corp, Kobe, Japan).

#### 4.4.2. Platelet Aggregation Assay

Platelet aggregation was conducted using a light transmission aggregometer (AggRAM^™^, Helena Biosciences, Gateshead, UK) [65,66]. PRP (150–600 × 10^9^ platelet/L) was stirred in an aggregometer cuvette at 37 °C, and different coffee extracts (0.5 mg/mL and 1 mg/mL, 25 µL) were added. After incubation for 15 min, 25 µL of PBS (control) or agonists including: ADP (1–10 µM), epinephrine (1–10 µM), collagen (1–4 µg/mL), TRAP (12.5 µM), ARA (1 mM) and U46619 (1 µg/mL) were added to the mixture, and it was further incubated for 5 min. Aggregation was monitored as light transmission at 650 nm for 300 sec. The percentage of platelet aggregation inhibition was calculated using the following formula; 100 × (OD_PBS_ − OD_PRP_)/OD_PBS_.

In a time-course study, PRP was incubated with different coffee extracts (1 mg/mL each, 25 μL) in an aggregometer cuvette at 37 °C for 15, 30, and 60 min. Finally, PBS or agonists including 1 μg/mL collagen and 1 mM ARA (25 μL) were added to the mixture and platelet aggregation was monitored as has been mentioned above. The percentage of platelet aggregation inhibition was calculated using the following formula: 100 × (OD_PBS_ − OD_PRP_)/OD_PBS_.

### 4.5. Determination of Cyclooxygenase Activity

Biologically, COXs (or PGH synthase) that exhibit both COX and peroxidase (POD) activities catalyze the conversion of ARA to hydroperoxy endoperoxide (or PGG_2_), which will subsequently be reduced by the POD component to PGH_2_, a precursor of PGs, thromboxane and prostacyclins. The COX inhibitor screening method is based on the measurement of PGF_2α_ by the stannous chloride (SnCl_2_)-catalyzed reduction of PGH_2_ that is produced in the COX reaction via cELISA using a broadly specific antibody against all PGs and Ellman’s reagent [67]. The assay reagent includes both ovine COX-1 and human recombinant COX-2 enzymes together with acetylcholinesterase-conjugated PG (AChE-PG) tracer and Ellman’s reagent containing acetylcholine and 5,5′-dithio-bis-(2-nitrobenzoic acid). In this study, five different extracts of espresso, boiled and drip coffee (10 μL each) in duplicate were firstly incubated with the COXs reagent (10 μL) at 37 °C for 10 min. Secondly, the 10 μM ARA substrate solution (10 μL) was added to the mixture, and the mixture was then further incubated for exactly 2 min. Thirdly, SnCl_2_ solution (30 μL) was added to the mixture to stop the process of enzyme catalysis. Fourthly, treated coffee extracts, standard PG (15.6–2000 pg/mL) and ELISA buffer (50 μL each) were added to mouse anti-rabbit IgG-immobilized plate wells, followed by the addition of AChE-PG tracer (50 μL) to complete the free PGs for the purpose of binding the rabbit antibody against PGs (50 μL), for which the AChE-PG-anti-PG complex was subsequently bound to the immobilized anti-rabbit IgG. The amount of AChE-PG that could bind to the anti-PG was found to be inversely proportional to the concentrations of PGs in the well. Ellman’s reagent was added into each well to develop the yellow colored 5-thio-2-nitrobenzoic acid product that was determined photometrically at 405 nm using a microplate reader. The percentage of COX activity inhibition was obtained using the following forula: 100 × (OD_PBS_ − OD_PRP_)/OD_PBS_.

### 4.6. Statistical Analysis

Experimental data were analyzed using SPSS Statistics Program (IBM SPSS^®^ Software version 22, IBM Corporation, Armonk, NY, USA, shared license by Chiang Mai University, Chiang Mai, Thailand). Data are expressed as mean ± standard deviation (SD) values with standard error of mean (SEM). Statistical significance was determined using one-way analysis of variance (ANOVA) followed by Tukey’s HSD posttest. *p* value < 0.05 was considered statistically significant.

## 5. Conclusions

Limitations of the study include the fact that only small numbers of subjects were studied; that a small range of sources of coffee product (e.g., Arabica coffee from the Royal Project Foundation) was used and the fact that only in vitro studies were performed and it is therefore not certain whether the same effect would be observed ex vivo in normal volunteers. Overall, the results strongly corroborate the hypothesis that the brewing procedure, type of coffee and potential active compounds, such as caffeine, chlorogenic acid and phenolic compounds, could influence platelet aggregation and the activity of cyclooxygenases. Notably, drip and espresso coffee brews are more effective in amelioration of cyclooxygenase-catalyzed anti-/pro-inflammation. Major bioactive compounds in the coffee beverages need to be identified using liquid chromatography/mass spectrometry together with a thorough assessment of their potential anti-thrombotic effects. Consequently, coffee beverages might indeed lower the risk of developing a hypercoagulable state and thrombosis.

## Figures and Tables

**Figure 1 molecules-26-00010-f001:**
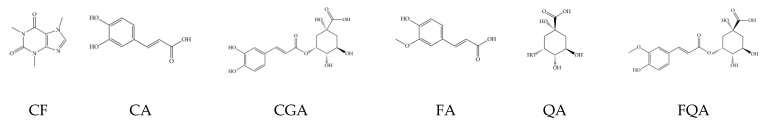
Chemical structures of caffeine (CF), caffeic acid (CA), chlorogenic acid (CGA), ferulic acid (FA), quinic acid (QA), and feruloylquinic acid (FQA) present in coffee [2].

**Figure 2 molecules-26-00010-f002:**
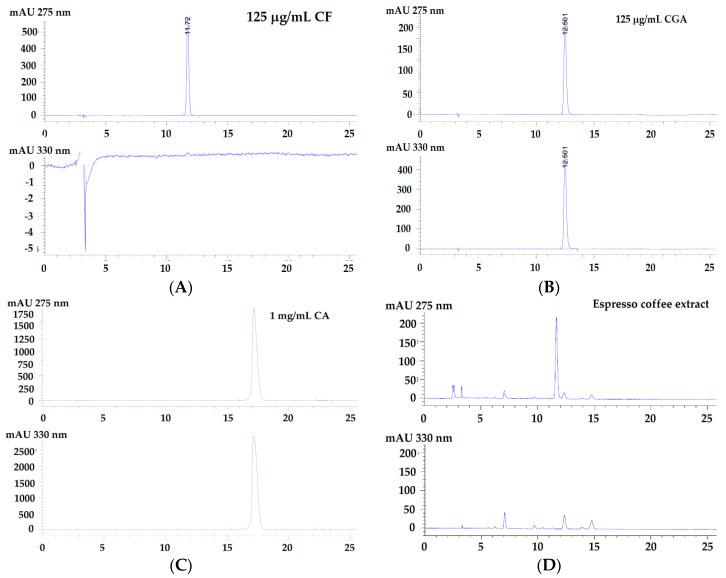
HPLC/DAD profiles of authentic standards including caffeic acid (CA), caffeine (CF), and chlorogenic acid (CGA); espresso, drip, boiled and instant coffee extracts (**A**–**H**).

**Figure 3 molecules-26-00010-f003:**
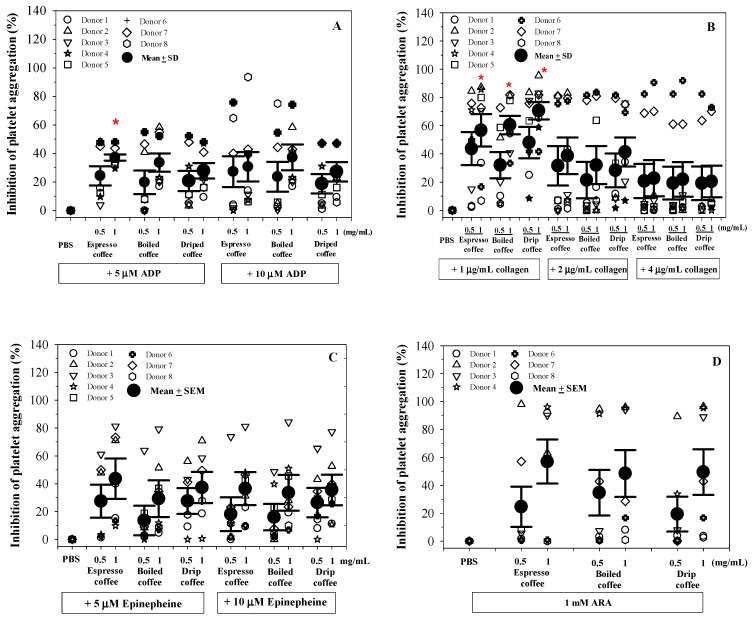
Inhibitory effects of phosphate-buffered saline (PBS) and coffee extracts (0.5 and 1 mg/mL each) on aggregation of platelets induced by adenosine diphosphate (ADP) (**A**), collagen (**B**), epinephrine (**C**), sodium arachidonate (ARA) (**D**), U46619 (**E**) and thrombin receptor agonist peptide (TRAP) (**F**). Data obtained from different blood samples are expressed as mean ± SEM values. * *p* < 0.05 when compared without the coffee extract treatments (PBS).

**Figure 4 molecules-26-00010-f004:**
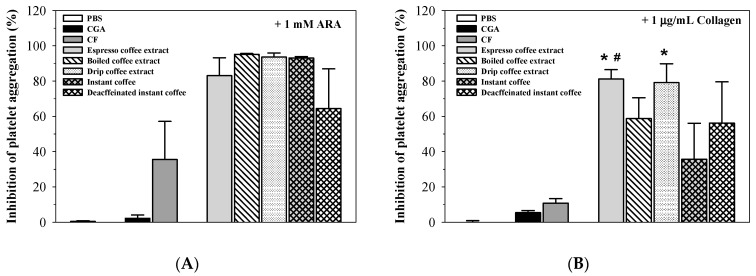
Inhibitory effects of phosphate-buffered saline (PBS), coffee extracts (0.5 and 1 mg/mL each), 18.23 mg/mL chlorogenic acid (CGA) and 54.17 mg/mL caffeine (CF) on aggregation of platelets induced by collagen (**B**) and sodium arachidonate (ARA) (**A**). Data obtained from five blood samples are expressed as mean + SEM values. * *p* < 0.05 when compared with the CGA; # *p* < 0.05 when compared to the CF.

**Figure 5 molecules-26-00010-f005:**
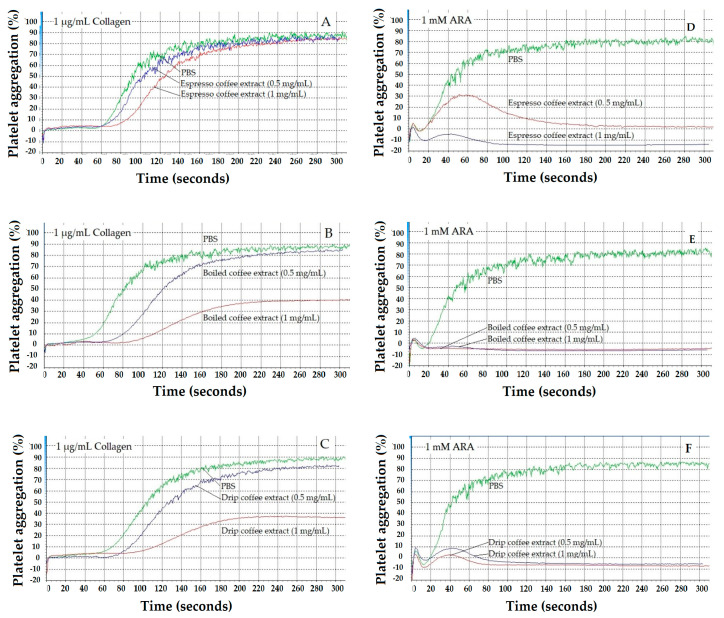
Representative aggregation traces of collagen (**A**–**C**) and sodium arachidonate (ARA)(**D**–**F**)-induced platelet aggregation by espresso, boiled, and drip coffee extracts (1 mg/mL each). Data were obtained from three platelet-rich plasma samples and are presented as mean ± SEM values.

**Figure 6 molecules-26-00010-f006:**
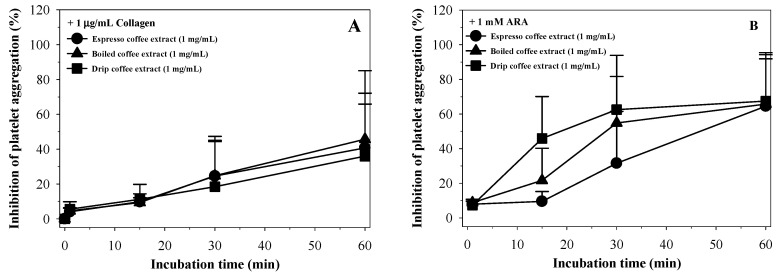
Time-course experiments of collagen (**A**) and sodium arachidonate (ARA) (**B**)-induced platelet aggregation by espresso, boiled and drip coffee extracts (1 mg/mL each). Data were obtained from three platelet-rich plasma samples and are presented as mean ± SEM values.

**Figure 7 molecules-26-00010-f007:**
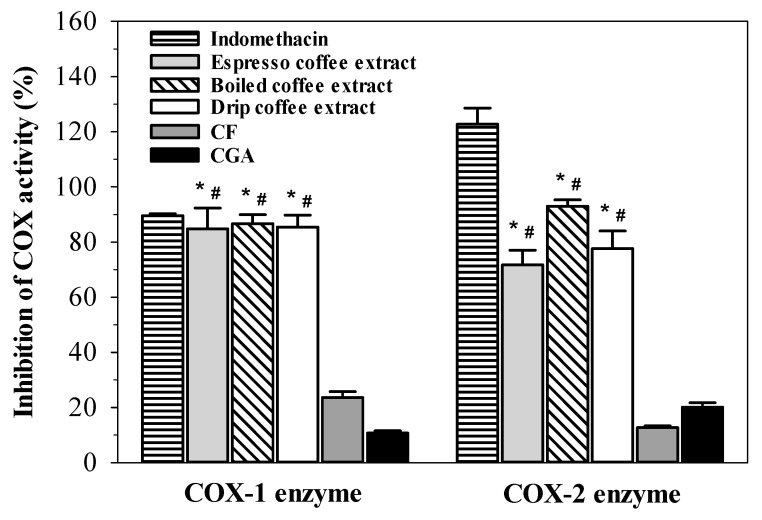
Inhibition of cyclooxygenase 1 (COX-1) and cyclooxygenase 2 (COX-2) activities by indomethacin (100 mg/mL), caffeine (CF, 100 μg/mL), chlorogenic acid (CGA, 20 μg/mL), espresso coffee, boiled coffee, and drip coffee extracts (2 mg/mL each). Data obtained from five separate duplicate measurements are expressed as mean ± SEM values. * *p* < 0.05 when compared with the CGA; # *p* < 0.05 when compared to the CF.

**Figure 8 molecules-26-00010-f008:**
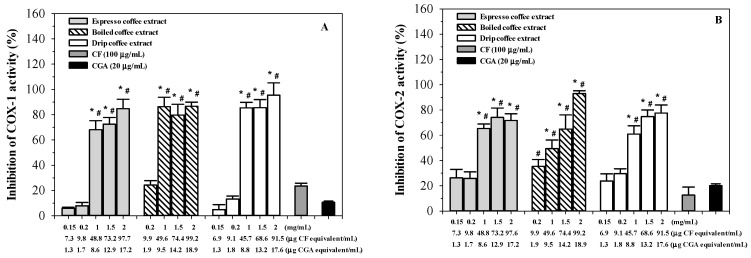
Inhibitory effects of espresso, boiled and drip coffee extracts (0.15–2 mg/mL each), caffeine 100 μg/mL) and chlorogenic acid (CGA, 20 μg/mL) on cyclooxygenase 1 (COX-1) (**A**) and cyclooxygenase 2 (COX-2) (**B**) activities. Data obtained from five separate duplicate measurements are expressed as mean ± SEM values. * *p* < 0.05 when compared with the CGA; # *p* < 0.05 when compared to the CF.

**Table 1 molecules-26-00010-t001:** Average values of chlorogenic acid, caffeine, total phenolic contents and antioxidant activity in coffee extracts.

Extract	CGA (μg/mg)	CF (μg/mg)	CA (μg/mg)	TPC (mg GAE/g)	Antioxidant Activity (mg TE/g)
**Espresso Coffee**	8.60	48.83	ND	113	125
**Drip Coffee**	8.81	45.73	ND	104	160
**Boiled Coffee**	9.45	49.58	ND	114	118
**Instant (Regular) Coffee**	5.15	39.81	ND	135	156
**Instant (Decaffeinate) Coffee**	10.61	0.53	ND	166	163

Abbreviations: CA = caffeic acid, CF = caffeine, CGA = chlorogenic acid, GAE = gallic acid equivalent, ND = not detectable, TE = Trolox equivalent, TPC = total phenolic content.

## Data Availability

Data available in a publicly accessible repository.

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
