# Peer review of "Anti-Platelet Aggregation and Anti-Cyclooxygenase Activities for a Range of Coffee Extracts (Coffea arabica)"

_molecules, 2020, doi:10.3390/molecules26010010_

Round 1

Reviewer 1 Report

Figure 3 – spelling espresso on the ADP panel A

Figure 3 the authors state in the text a significant change is observed with ARA but no significance values are shown on the graph for ARA.  Please include or amend the text accordingly.

Line 189 – should be significant not significantly.

Author Response

Department of Biochemistry

Faculty of Medicine

Chiang Mai University

Chiang Mai, Thailand

December 16th, 2020

Re: Minor revision of the manuscript (ID: Molecules-1035240)

Dear Assistant Editor of Molecules journal (Ms. Lana Dzodanovic),

I am enclosing here with the revised manuscript (ID: Molecules 1035240) R1 entitled “Anti-Platelet Aggregation and Anti-Cyclooxygenase Activities for a Range of Coffee Extracts (Coffea arabica)” to be considered and published in your esteem journal. We have revised the manuscript according to the editor and two reviewers’ comments point by point and track changes as shown in highlight colour. Responses to all the comments have been reported. 

The article contains abstract (200 words), main text (19 pages) included 67 references, 1 table and 8 figures included a modified Figure 3A. All the authors have read and approved of the revised manuscript. We declare there is no conflict of interest regarding the publication of this article.

Yours sincerely,

Professor Somdet Srichairatanakool, Ph.D.                                                          Corresponding author                                                                                        Email: somdet.s@cmu.ac.th

Reviewer 2 Report

I believe that the authors have addressed the previous concerns, and that the article is now acceptable for publication in Molecules

Author Response

Department of Biochemistry

Faculty of Medicine

Chiang Mai University,

Chiang Mai, Thailand

December 16th, 2020

Re: Minor revision of the manuscript (ID: Molecules-1035240)

Dear Assistant Editor of Molecules journal (Ms. Lana Dzodanovic),

I am enclosing here with the revised manuscript (ID: Molecules 1035240) R1 entitled “Anti-Platelet Aggregation and Anti-Cyclooxygenase Activities for a Range of Coffee Extracts (Coffea arabica)” to be considered and published in your esteem journal. We have revised the manuscript according to the editor and two reviewers’ comments point by point and track changes as shown in highlight colour. Responses to all the comments have been reported. 

The article contains abstract (200 words), main text (19 pages) included 67 references, 1 table and 8 figures included a modified Figure 3A. All the authors have read and approved of the revised manuscript. We declare there is no conflict of interest regarding the publication of this article.

Yours sincerely,

Professor Somdet Srichairatanakool, Ph.D.                                                          Corresponding author                                                                                        Email: somdet.s@cmu.ac.th

This manuscript is a resubmission of an earlier submission. The following is a list of the peer review reports and author responses from that submission.

Round 1

Reviewer 1 Report

The work presented here evaluates the composition of various classes of coffee extract, and then evaluates the anti-inflammatory and anti-oxidant properties of each extract, as well as the properties of some of the isolated components.

An interesting overview of the health benefits and drawbacks of coffee drinking is also presented.

While this work is not particularly ground-breaking, the research is clearly presented, logically-planned and well-written. It would be interesting to see if these in vitro results would be retained in an in vivo scenario, a limitation which the authors themselves state within the paper (indeed, the authors provide an honest declaration of the limitations of the study at the end of the manuscript). 

Overall, I would recommend this manuscript for publication in the journal 'Molecules' after the following minor alterations have been made:

  • Readers (particularly chemists) would beneift from a Figure within the main manuscript showing the chemical structures of caffeine, caffeic acid, chlorogenic acid, feruloylic acid, quinic acid, feruloylquinic acid, caffeoylquinic acid. With the exception of caffeine, these structures are somewhat related to each other, and this significance is not conveyed to the reader in the present form.
  • The abbreviation for caffeic acid (CA) is first defined on Line 85. However 'CA' is first referred to on Line 44, meaning the abbreviation definition should be placed earlier. 
  • Line 384: Should be a gap between value and units in '170g'.
  • On Lines 51/52, chlorogenic acid (CGA) is referred to a nitrogeneous compound. However, CGA does not contain nitrogen, and therefore the word nitrogeneous this should be removed.

Author Response

The work presented here evaluates the composition of various classes of coffee extract, and then evaluates the anti- inflammatory and anti-oxidant properties of each extract, as well as the properties of some of the isolated components. An interesting overview of the health benefits and drawbacks of coffee drinking is also presented.

While this work is not particularly ground-breaking, the research is clearly presented, logically-planned and well-written. It would be interesting to see if these in vitro results would be retained in an in vivo scenario, a limitation which the authors themselves state within the paper (indeed, the authors provide an honest declaration of the limitations  of  the study at the end of the manuscript).

Overall, I would recommend this manuscript for publication in the journal 'Molecules' after the following minor  alterations have been made:

Point 1: Readers (particularly chemists) would benefit from a Figure within the main manuscript showing the chemical structures of caffeine, caffeic acid, chlorogenic acid, feruloylic acid, quinic acid, feruloylquinic acid, caffeoylquinic acid.

With the exception of caffeine, these structures are somewhat related to each other, and this significance is not conveyed to the reader in the present form.

Response1:

Page 1, Line 39-40: The words “…, caffeic acid (CA), ferulic acid (FA), quinic acid (QA) and feruloylquinic acid (FQA) (Figure 1), ..” have been added

Page 2: We have drawn and added chemical structures of caffeine, caffeic acid, chlorogenic acid, feruloylic acid, quinic acid, feruloylquinic acid, caffeoylquinic acid in Figure 1

Page 2, Line 44-45: Figure 1 Chemical structures of caffeine (CF), caffeic acid (CA), chlorogenic acid (CGA), ferulic acid (FA), quinic acid (QA) and feruloylquinic acid (FQA) found in coffee [2].” has been added

Point 2: The abbreviation for caffeic acid (CA) is first defined on Line 85. However 'CA' is first referred to on Line 44, meaning the abbreviation definition should be placed earlier.

Response 2: Thank you very much. We have corrected them.

Page 1, Line 39: The phase “..5-O-caffeoylquinic acid (CQA) and quinic acid (QA),” has been modified to be “…5-O- caffeoylquinic acid (CQA), caffeic acid (CA), ferulic acid (FA), quinic acid (QA) and feruloylquinic acid (FQA) (Figure 1),” Page 1, Line 40: Figure 1 together with the chemical structures has been added between “…effects [2].” and “Boiled coffee…”. Consequently, Figures 1-7 on Page 4, 5, 6, 6-7, 8, 9 and 10 have been renumbered as Figure 2-8, respectively.

Page 1, Line 44: Now, “caffeic acid (CA)” has been first defined on Page 1, Line 39 and the word “CA” can be use later on Page 1,Line 44.

Page 2, Line 85: “…caffeic acid (CA)…” has been replaced by “…CA…” Page 11, Line 224: “ferulic acid (FA)” has been changed to be “FA”   Page 11, Line 229: “quinic acid” has been changed to be “QA”

Point 3: Line 384: Should be a gap between value and units in '170g'.

Response 3: I accept the mistake and have replaced “170g” with “170 g”

Point 4: On Lines 51-52, chlorogenic acid (CGA) is referred to a nitrogeneous compound. However, CGA does not contain nitrogen, and therefore the word nitrogeneous this should be removed.

Response 4: Page 2, Line 60-61: The phase “…by the nitrogenous compounds CF and CGA [3,7]” has been rewritten as “…by the compounds such as CF and CGA [3,7]”.

Reviewer 2 Report

In this manuscript the Authors determine the chemical composition of coffee prepared by various methods. They investigate the effect of the coffee extracts on the platelet aggregation and COX activity after stimulation with multiple platelet agonists. The authors show that coffee extracts have the most profound effect on platelet aggregation induced by arachidonic acid. I have a number of concerns as detailed below.

Did you control for coffee/caffeine consumption of their blood volunteers? This is important for interpretation of the data and could account for variability in the aggregation response.  

How does the doses of extracts tested (0.5 and 1 mg/ml) relate to those achieved after consumption of coffee? Are these physiologically relevant concentrations?

Figure 2 – The legend states that 1 mg/ml coffee extract was used but my understanding from the graphs are that both 0.5 and 1 mg/ml are used. There looks to be quite large changes in each panel of this figure yet very few are shown to be statistically significant, some of which are surprising. I think it would be useful to show the individual points rather than a bar chart to show the spread in the data. It also suggested that the sample size may be too small.

Figure 3A shows no difference between the coffee extracts I don’t think an order of effect is established as the authors suggested.

The authors state that Espresso slowly inhibited platelet aggregation. However, the rate of aggregation was slightly slower but the overall response (max aggregation) was not altered. Likewise, the lower concentration (0.5 mg/ml) of boiled or drip coffee did not greatly alter the max aggregation achieved.

How was the data in Figure 4 D calculated? Is the time shown a preincubation time or should the x-axis be in seconds as described in the methods aggregation was monitored for 300 s? From figure 4 D there looks to be no difference between the preparations as stated by the authors.

The traces show that boiled coffee completely inhibited platelet aggregation, yet in figure 5D the average appears to be around 70 %. These traces also show a small amount of aggregation before disaggregating and the trace falling back to baseline can the authors explain this?

CGA and CF show a small effect on COX activity and platelet aggregation, even when used at considerably higher concentration than those that are obtained in the coffee extracts. This would suggest that another substance in the coffee is primarily causing the effect on platelet reactivity.  What happens if both CGA and CF are used, do they have a synergistic effect?

Minor comments:

Increase font size on labels and axis text on fig 1

I am not clear what the authors mean by ‘the peaks eluted between 2-10 min were observed to be evenly predominant’.

Figure 4 unable to read axis clearly.

Reviewer 3 Report

Authors have presented comparison of a range of coffee extracts with regard to platelet aggregation and anti-cyclooxygenase activity. Even though the study is original and is devoted to an interesting topic, some concerns must be addressed.

The description of assays for COX-1 and COX-2 activities is insufficient. Authors should clarify whether they were working with isolated recombinant COX-1 and COX-2 enzymes, cell cultures or some tissue extracts. In each variant there are some issues to be addressed. In case if experiments are carried out with recombinant proteins in cell free system, authors should explain how they could determine the formation of PGF2alpha, since it is not a direct product by cyclooxygenases. The only direct product of COX-1 and COX-2 is an unstable intermediate PGH2, which is converted to prostaglandins by corresponding prostaglandin synthases. PGF2alpha could be synthesized either from PGH by aldo-keto reductase or from PGE2. In case if experiments are carried out in cell cultures or extracts, it should  explained in details. How could inhibition of COX-2 by indomethacyne be more than 100% (Figure 6)? How effects on COX-1 and COX-2 activities were separated, if it was not done using recombinant proteins? This confusion could be resolved by more detailed description of assays.

More experiments are probably necessary for statistically significant results (two measurements only are  done in studying cyclooxygenase activity).

Round 2

Reviewer 2 Report

The authors have made substantial clarifications to the text and have now included the relevant information to show that they have controlled for caffeine intake. 

The spread of the data can now be seen in Figure 3, the authors state themselves in the response that the sample size is too small to see an effect due the donor variability.  Is it possible to Increase the sample size which may help to increase the statistical power of some of the trends observed? 

There are no apparent differences between the extracts except for perhaps decaf coffee (if this is statistically significant) in Figure 4 with ARA.  I still think it would simplify the message to state there was no difference. 

Author Response

The authors have made substantial clarifications to the text and have now included the relevant information to show that they have controlled for caffeine intake.

Point 1: Page 6, Figure 3

The spread of the data can now be seen in Figure 3, the authors state themselves in the response that the sample size is too small to see an effect due the donor variability.  Is it possible to Increase the sample size which may help to increase the statistical power of some of the trends observed? 

Answer:

We have now added the results of platelet aggregation inhibitions from three more blood samples (Donors 5-8) that were induced by ARA, ADP, collagen and epinephrine to the original results (Donors 1-4). Additionally, we have plotted new graphs in Figure 3 and analysed the statistical significance of all of the results.

Pages 4-5, Lines 119-129 2.2. Inhibitory Effect of Coffee Extracts on Platelet Aggregation

Answer:

Page 5, Line 124-128

We have agreed with the reviewer and edited text accordingly. The sentences have been rewritten as follow: “…showed the significant inhibition. However, the inhibitory effects became weaker when the doses of the agonists were increased. The most marked inhibition was seen with low doses of collagen and with ARA. In addition, the inhibitory effects of the coffee extracts on the platelet aggregation to U46619 were weaker (only occurring in 1/5 donors) and there was no inhibition of aggregation to TRAP (Figure 3).”

Page 11, Line 287-291

We have also edited text accordingly. The sentences have been rewritten as follow: “…collagen, and ARA, for which the most remarkable degree of inhibition took place with the ARA agonist, while inhibition was more marked at lower doses of collagen (which are thought to activate platelets mainly through the cyclooxygenase pathway).

Point 2: Page 7, Line 145-146

There are no apparent differences between the extracts except for perhaps decaffeinate coffee (if this is statistically significant) in Figure 4 with ARA.  I still think it would simplify the message to state there was no difference. 

Answer: Page 7, Line 146-156

The sentences have been now rewritten as follow: “…There were no differences in the degree of inhibition between the coffee types at equal concentrations. Similarly, CF and CGA slightly inhibited the aggregation of platelets induced by collagen and showed much lower inhibitory effects than the coffee extracts. Therefore, the inhibitory effects of Arabica coffee and instant coffee on platelet aggregation were interesting, but will require further investigation.” 

Reviewer 3 Report

Authors addressed this reviewer concerns in revised manuscript.
